# Identification of C_21_ Steroidal Glycosides from *Gymnema sylvestre* (Retz.) and Evaluation of Their Glucose Uptake Activities

**DOI:** 10.3390/molecules26216549

**Published:** 2021-10-29

**Authors:** Meiyu Liu, Tongxi Zhou, Jinyan Zhang, Guangfeng Liao, Rumei Lu, Xinzhou Yang

**Affiliations:** 1School of Pharmaceutical Sciences, Guangxi University of Chinese Medicine, Nanning 530200, China; lmy13263808041@163.com (M.L.); 18435166743@163.com (J.Z.); guangfengliao@126.com (G.L.); 2School of Pharmaceutical Sciences, South-Central University for Nationalities, Wuhan 430074, China; tc13627123095@163.com

**Keywords:** *Gymnema sylvestre*, C_21_ steroidal glycosides, glucose uptake, GLUT-4

## Abstract

*Gymnema sylvestre* (Retz.) Schult is a multi-purpose traditional medicine that has long been used for the treatment of various diseases. To discover the potential bioactive composition of *G. sylvestre*, a chemical investigation was thus performed. In this research, four new C_21_ steroidal glycosides sylvepregosides A-D (**1**–**4**) were isolated along with four known compounds, gymnepregoside H (**5**), deacetylkidjoladinin (**6**), gymnepregoside G (**7**) and gymnepregoside I (**8**), from the ethyl acetate fraction of *G. sylvestre*. The structures of the new compounds were established by extensive 1D and 2D nuclear magnetic resonance (NMR) spectra with mass spectroscopy data. Compounds **1**–**6** promoted glucose uptake by the range of 1.10- to 2.37-fold, respectively. Compound **1** showed the most potent glucose uptake, with 1.37-fold enhancement. Further study showed that compounds **1** and **5** could promote GLUT-4 fusion with the plasma membrane in L6 cells. The result attained in this study indicated that the separation and characterization of these compounds play an important role in the research and development of new anti-diabetic drugs and pharmaceutical industry.

## 1. Introduction

Diabetes mellitus (DM) is a disease caused by lack of insulin secretion or insulin resistance, which cause the body to experience persistent hyperglycemia and long-term metabolic disorders. DM is classified into type 1 diabetes mellitus (T1DM) and type 2 diabetes mellitus (T2DM), in which T2DM accounts for nearly 95% of individuals, and the number of patients is increasing year by year [1,2]. Insulin resistance (IR) is one of the essential conditions in the development of type 2 diabetes mellitus, and insulin reduces blood sugar levels and improves IR by promoting glucose absorption [3]. Traditional Chinese medicine has used natural products (NPs), including TCM formulations and their extracts, to treat human diseases for thousands of years. Many Chinese herbal medicines and their active ingredients have anti-diabetic properties with minimal side effects, and are widely used in the treatment of T2DM [4]. Therefore, it is of great significance to screen new effective hypoglycemic drugs from natural products through glucose uptake.

*Gymnema sylvestre* (Retz.) Schult is a genus of tropical plants belonging family Apocynaceae, and mainly distributed in Guangxi, Guangdong, Yunnan, Fujian and Zhejiang provinces of China [5]. *G. sylvestre* was used as a natural remedy for over two millennia. As a traditional Chinese herbal medicine, *G. sylvestre* has the effects of detumescence, fever removal, detoxification, promoting muscle growth, reducing swelling, dispelling wind and relieving pain, and is used to treat diseases such as vasculitis, snake bites, rheumatism, waist and knee pain, etc. [6,7]. Furthermore, *G. sylvestre* has also demonstrated other important uses, such as hypolipidemic, antiviral, diuretic, antiallergic and antibiotic uses, as well as use as a weight loss supplement [8]. In addition, *G. sylvestre* is also a traditional Chinese ethnic medicine used by Zhuang, Yao, Miao and other ethnic minorities for a long time. Zhuang medicinal ancient books has recorded that *G. sylvestre* has obvious effects in treating diabetes [9]. *G*. *sylvestre* is widely used as a naturopathic treatment for diabetes. Modern pharmacological studies have found that *G. sylvestre* leaf, root and stem extracts all have hypoglycemic effect [10,11,12]. Previous studies of this plant afforded that it has broad biological activities for anti-diabetic effect, for example, by suppressing elevated glucose levels in blood, recovering pancreatic beta cells and increasing insulin sensitivity to treat type 2 diabetes mellitus in the earliest stage of disease [13,14,15,16]. The serum levels of GLUT-4, PPAR-γ, adiponectin, and leptin have a positive correlation with insulin sensitivity. Study shows that methanolic leaf extract of *G. sylvestre* (MLGS) could upregulate the expression of GLUT-4, PPAR-γ, adiponectin, and leptin, thereby exerting a potent insulin-sensitizing effect [17]. In recent studies, it was also shown that gymnemic acids (GA) separated from *G. sylvestre* (Retz.) Schult decreased blood lipid levels and increased insulin activity in experiments using murine models [18]. The hypoglycemic mechanisms of GA may be related to promoting insulin signal transduction and activating PI3K/Akt- and AMPK-mediated signaling pathways in T2DM rats [19]. Apart from these, GA has the potential for inhibiting electrogenic glucose uptake in the gastrointestinal tract by inhibiting sodium-dependent glucose transporter 1 [20]. Conduritol A is also considered to be the active ingredient in hypoglycemic activity of *G. sylvestre*. Experimental results in rats showed that conduritol A can significantly reduce blood glucose levels and increase blood insulin levels. The hypoglycemic mechanism may be related to improving insulin sensitivity, scavenging free radicals, promoting liver glycogen synthesis, enhancing antioxidant capacity and enhancing immune function [21]. The main components found in *G. sylvestre* are saponins, polysaccharides, sterols, terpenoids, flavonoids, peptides, pectin and so on [22]. Aiming to enhance understanding of the chemical and more anti-diabetic activities constituent of *G. sylvestre*, we performed purification of the EtOAc fraction of the traditional herb, which led to four new C_21_ steroidal glycosides sylvepregosides A-D (**1**–**4**), together with four known: gymnepregoside H (**5**), deacetylkidjoladinin (**6**), gymnepregoside G (**7**) and gymnepregoside I (**8**) (Figure 1). In addition, compounds **1**–**6** have effects on increasing glucose uptake in vitro. Compounds **1** and **5** showed a strong glucose uptake in L6 cells, with enhancements up to 2.37- and 1.96-fold. Herein we described the isolation, structure elucidation, the glucose uptake activities as well as the GLUT-4 fusion with the plasma membrane of these isolated compounds.

## 2. Results and Discussion

Compound **1** was isolated as a white powder. The molecular formula was established as C_51_H_68_O_15_ by HRESIMS data (*m/z* 943.4423 [M + Na]^+^, calcd for 943.4450) (Appendix A). The IR spectrum (Appendix A) showed the characteristic of hydroxyl (3480 cm^−1^), ester (1713 and 1281 cm^−1^), and olefinic (1640 cm^−1^) groups. The ^1^H and ^13^C NMR spectra (Table 1) of **1** showed the characteristic signals for the aglycone of C_21_ steroidal glycosides isolated from *Gymnema aliemifoliu* [23], with two angular methyl groups CH_3_-18 and CH_3_-19 (*δ*_H_ 1.60, 1.01; *δ*_C_ 11.5, 21.1), a secondary methyl group CH_3_-21 (*δ*_H_ 1.35; *δ*_C_ 15.4), a double bond (*δ*_H-6_ 5.63 and *δ*_H-7_ 5.83; *δ*_C_ 136.7, 126.3), three oxygenated tertiary carbons, CH-3, CH-12, and CH-20 (*δ*_H_ 4.07~4.10, 4.74~4.77, and 4.84; *δ*_C_ 74.9, 74.8, and 74.7), and five quaternary carbons, C-5, C-8, C-10, C-14, and C-17 (*δ*_C_ 74.6, 74.3, 39.2, 87.5 and 87.8). The above signals the establishment of the 3,5,8,12,14,17,20-heptahydroxypregnane skeleton [24,25], which was further confirmed by detailed analyses of its HSQC, ^1^H-^1^H COSY and HMBC spectra (Appendix A). In addition, a cinnamoyl group [*δ*_H_ 6.07 and 7.42 (eatch 1H, d, *J* = 16.0 Hz, H-2’ and H-3’), *δ*_H_ 7.24 (2H, d, *J* = 7.3 Hz, H-5′,9′), 7.30~7.36 (5H, m, H-6′, 7′, 8′); *δ*_C_ 166.6, 118.9, 144.3, 134.4, 128.3, 128.5, and 130.2] and a *O*-benzoyl group [*δ*_H_ 7.92 (2H, d, *J* = 7.2 Hz, H-3″, 7″), 7.30~7.36 (5H, m, H-4″, 6″) and 7.53 (1H, t, *J* = 7.4 Hz, H-5″); *δ*_C_ 165.0, 130.4, 129.9, 128.7 and 133.1] were observed in ^1^H and ^13^C NMR spectrum. The HMBC correlations between H-12 (*δ*_H_ 4.74~4.77) and C-1′ (*δ*_C_ 166.6); between H-20 (*δ*_H_ 4.84) and C-1″ (*δ*_C_ 165.0) revealed the placement of the two ester groups at C-12 and C-20. The ^1^H NMR spectrum of **1** showed two anomeric proton signals at *δ*_H_ 4.81 (1H, dd, *J* = 9.7, 1.6 Hz) and 4.67 (1H, dd, *J* = 9.7, 1.6 Hz), corresponding to carbon resonances at *δ*_C_ 97.7, and 99.5 confirmed by HSQC, and the coupling constants of anomeric proton revealed the sugars were *β*-linkage. The structures of saccharides were determined by analysis of NMR data, including HSQC, ^1^H-^1^H COSY, HMBC, NOESY and 1D/2D TOCSY experiments (Appendix A), and further confirmed by comparison of the data with those in the literature [26]. Thus two monosaccharide units were both characterized as *β*-cymarose. Taking *β*-cymaroseII of **1** as an example, in the 1D-TOCSY experiment, irradiation of the signal at *δ*_H_ 3.62 (H_cymII_-3, 1H, dd, *J* = 6.0, 2.9 Hz) enabled identification of H_cymII_-1 (*δ*_H_ 4.67, 1H, dd, *J* = 9.7, 1.6 Hz), H_cymII_-2a (*δ*_H_ 1.62~1.64, 1H, m), H_cymII_-2b (*δ*_H_ 2.23~2.27, 1H, m), H_cymII_-4 (*δ*_H_ 3.19, 1H, dd, *J* =9.7, 2.8 Hz) and H_cymII_-6 (*δ*_H_ 1.27, 1H, d, *J* = 6.2 Hz) in the same conjugation system. According to the coupling constant of each proton signal, the kind of saccharide moiety could be identified. These proton signals were further assigned based on analysis of its ^1^H-^1^H COSY spectrum [^1^H-^1^H COSY correlations between H_cymII_-1 (*δ*_H_ 4.67) and H_cymII_-2a (*δ*_H_ 1.62~1.64), H_cymII_-2a (*δ*_H_ 1.62~1.64) and H_cymII_-3 (*δ*_H_ 3.62), H_cymII_-3 (*δ*_H_ 3.62) and H_cymII_-4 (*δ*_H_ 3.19), H_cymII_-4 (*δ*_H_ 3.19) and H_cymII_-5 (*δ*_H_ 3.55), H_cymII_-5 (*δ*_H_ 3.55) and H_cymII_-6 (*δ*_H_ 1.27)]. The HMBC spectrum revealed ^1^H-^13^C long-range correlations of H_cymI_-1 to C-3 (*δ*_C_ 74.9); and H_cymII_-1 to C_cymI_-4 (*δ*_C_ 82.2), which illustrates the connection sequence of oligosaccharide chain and its location at C-3 of the aglycone in compound **1**. In 20-hydroxy-C/D-cis-pregnane-type steroids, steric hindrance between H-16, H-20, H-18 and H-21 of the cyclopentane ring can lead to chemical shifts and is indicative of the absolute configuration of the C-20 [27]. The NOE’s between H-9 (*δ*_H_ 1.87~1.90) and H-12 (*δ*_H_ 4.74~4.77), between H-18 (*δ*_H_ 1.60) and H-20 (*δ*_H_ 4.84), and our experimental data were similar to the closest structures published in the literature, which led to the deduction of the 20*S* configuration [28,29,30]. Thus, compound **1** was determined as 12-*O*-cinnamoyl-20-*O*-benzoyl-heptahydroxy-(20*S*)-pregn-6-enyl-3-*O*-*β* cymaropyranoside-(1→4)-*β-*cymaropyranoside.

Compound **2**, was isolated as a white powder. The molecular formula of **2** was determined as C_49_H_70_O_15_ by HRESIMS data (*m/z* 921.4578 [M + Na]^+^, calcd for 921.4607) (Appendix A). The NMR data (Table 1) of **2** were similar to those of **1**, except for the absence of an *O*-benzoyl group and the presence of a 2-methyl-2-butenoyl group [*δ*_H_ 6.72–6.77 (1H, m, H-3″), 1.68 (3H, d, *J* = 7.1 Hz, H-4″) and 1.71 (3H, s, H-5″); *δ*_C_ 166.1, 128.8, 138.1, 14.5 and 12.2]. The HMBC (Appendix A) correlations from H-20 (*δ_H_* 4.68) to C-1″ (*δ*_C_ 166.1) enabled location of the 2-methyl-2-butenoyl group at C-20. The aglycone of **2** was the same as stephanoside K isolated from *Stephanotis lutchuensis* [29]. Finally, the coupling constants of the anomeric proton *δ*_H_ 4.81 (1H, dd, *J* = 9.6, 1.8 Hz, H_cymI_-1) and 4.67 (1H, dd, *J* = 10.0, 2.3 Hz, H_cymII_-1) illustrated that the sugars were *β*-linkage. By comparison of the ^1^H and ^13^C NMR data for **2** with those of **1**, **2** was observed to have the similar aglycone and the same sugar sequence as those of **1**, and the key HMBC correlations confirmed that the sugar moiety was attached to C-3 of the aglycone (Figure 2). Thus, the structure of **2** was elucidated as 12-*O*-cinnamoyl-20-*O*-*(E)*-2-methyl-2-butenoyl-heptahydroxy-(20*S*)-pregn-6-enyl-3-*O*-*β*-cymaropyranoside-(1→4)-*β*-cymaropyranoside.

Compound **3** was isolated as a white powder. The molecular formula was established as C_42_H_66_O_14_ by HRESIMS data (*m/z* 817.4339, [M + Na]^+^, calcd for 817.4350) (Appendix A). The IR spectrum of **3** (Appendix A) showed the characteristic for hydroxy (3441 cm^−1^), ester (1701 cm^−1^) and olefinic (1681 cm^−1^) groups. By comparison of the NMR (Table 2) data of **3** with **1**, compound **3** show the similar skeleton with **1**, but the presence of the key HMBC (Appendix A) long-range correlations of *δ*_H_ 2.36~2.40 (H-4) to *δ*_C_ 118.4 (C-6), *δ*_H_ 5.36~5.38 (H-6) to *δ*_C_ 74.2 (C-8) and 37.1 (C-10), and *δ*_H_ 1.13 (H-19) to *δ*_C_ 139.9 (C-5) explained that the double bond was located at H-5 and H-6. In addition, the ^13^C NMR spectroscopic data of the skeleton of **3** matched well with those of the sarcostin skeleton isolated from *Cynanchum* botanicals [31,32]. By detailed analyses of its HSQC, ^1^H-^1^H COSY and HMBC spectra (Appendix A), a 2-methyl-2-butenoyl group [*δ*_H_ 6.83~6.87 (1H, m, H-3′), 1.82 (3H, d, *J* = 7.0 Hz, H-4′) and 1.85 (3H, s, H-5′); *δ*_C_ 166.4 (C-1′), 128.8 (C-2′), 138.0 (C-3′), 14.7 (C-4′) and 12.3 (C-5′)] and an acetyl group [*δ*_H_ 1.94 (3H, s, H-2″); *δ*_C_ 171.3 (C-1″) and 21.9 (C-2″)] were also observed. The key HMBC correlations of *δ*_H_ 4.65~4.66 (1H, m, H-20) to *δ*_C_ 166.4 (C-1′), and *δ*_H_ 4.62 (1H, dd, *J* = 11.3, 4.0 Hz, H-12) to *δ*_C_ 171.3 (C-1″) revealed the placement of the two ester groups at C-20 and C-12, respectively. The aglycone of **3** was the same as marstomentoside J isolated from *Marsdenia tomentosa* [33]. The ^1^H NMR spectrum of **3** showed two anomeric proton signals at *δ*_H_ 4.84 (1H, dd, *J* = 9.6, 1.7 Hz, H_cymI_-1) and 4.67 (1H, dd, *J* = 9.5, 1.8 Hz, H_cymII_-1), and the coupling constants of the anomeric protons illustrated that the sugars were *β*-linkage. By comparison of the ^1^H and ^13^C NMR data for **3** with those of **1**, **3** was observed to have the similar aglycone and the same sugar sequence as those of **1**, and the key HMBC correlations confirmed that the sugar moiety was also attached to C-3 of the aglycone (Figure 3). Based on the above evidence, the absolute configuration of **3** was elucidated as 12-*O*-acetyl-20-*O*-(*E*)-2-methyl-2-butenoyl-sarcostin-(20S)-3-*O*-β-cymaropyranoside-(1→4)-β-cymaropyranoside.

Compound **4** was isolated as a white powder. The molecular formula was established as C_35_H_54_O_11_ by HRESIMS data *m/z* 673.3553 [M + Na]^+^, (calcd for 673.3564) (Appendix A). The NMR data (Table 2) of **4** were very similar to those of **3**, except for the absence of one sugar group [*δ*_C_ 99.5, 33.8, 77.2, 82.6, 68.6, 18.3, 58.1]. The ^1^H NMR spectrum of **4** showed one anomeric proton signal at *δ*_H_ 4.78 (1H, dd, *J* = 9.6, 1.8 Hz), and the coupling constants of the anomeric proton illustrated that the sugar was *β*-linkage. In addition, the key HMBC (Appendix A) correlations confirmed that the sugar moiety was also attached to C-3 of the aglycone. Thus, the structure of **4** was characterized as 12-*O*-acetyl-20-*O*-(*E*)-2-methyl-2-butenoyl-sarcostin-(20*S*)-3-*O*-*β*-cymaropyranoside.

Four known compounds were also isolated and identified as gymnepregoside H (**5**), [23] deacetylkidjoladinin (**6**), [34] gymnepregoside G (**7**) [23] and gymnepregoside I (**8**) [35], by comparison of spectroscopic data and physicochemical properties with those reported values in the literatures.

Researches showed that berberine has been shown to have antidiabetic properties. Berberine could downregulated the expression of genes involved in lipogenesis and upregulated those involved in energy expenditure in adipose tissue and muscle [36,37]. The abundant compounds **1**–**6** (>2 mg) were selected to evaluate their glucose uptake activity in L6 cells. Berberine (BBR, 30 μg/mL) was used as the positive control, and we used a cell-based fluorescently-labeled deoxyglucose analog kit to test the uptake activity. After incubating with samples, the 2-NBDG fluorescence intensity at the plasma membrane shows varying degrees of change. Compared to normal control (NC) group, three new compounds sylvepregosides B-D (**2**–**4**) and deacetylkidjoladinin (**6**) exerted weak activity, and showed glucose uptake activity with the enhancement by 0.2-, 0.16-, 0.13- and 0.1-fold at 30 μg/mL (Figure 4). Gymnepregoside H (**5**) possesses a moderate effect on promoting glucose uptake, which increased glucose uptake to 1.96 folds at 30 μg/mL. Sylvepregosides A (**1**) was the most active compound, and exhibited glucose uptake activity with 1.37-fold enhancement at 30 μg/mL. Comparing **1** and **5**, both exhibited two groups of aromatic signals on C-12 and C-20, the only difference was the oligosaccharide chain in compound **1**, whereby the polysaccharide may enhance the activities of glucose uptake.

For further evaluation of the efficacy of **1** and **5**, a L6 cell line which stably expressed Myc-GLUT4-mOrange was used to test whether samples could also facilitate the fusion of GLUT4 with the plasma membrane. The red fluorescence of GLUT4-mOrange and the green fluorescence of FITC-myc were detected using laser confocal microscopy. As a result, we found that gymnepregoside H (**5**) possesses a moderate effect on promoting GLUT-4 fusion with the plasma membrane in L6 cells. Sylvepregosides A (**1**) was the most active compound promoting GLUT-4 fusion with the plasma membrane in L6 cells (Figure 5). Quantification of this effect revealed that the percentage of FITC positive cells were 69% and 65%, respectively (Figure 6).

## 3. Materials and Methods

### 3.1. Chemicals and Reagents

Chromatography grade solvents were used for HPLC, and all other chemical reagents were analytical grade. HPLC grade acetonitrile and methanol were purchased from Merck (Darmstadt, Germany). Sephadex LH-20 dextran gel was purchased from Amersham Pharmacia Biotech Co. (Piscataway, NJ, USA).

### 3.2. General Experimental Procedures

Semi-preparative HPLC purification was performed on a Waters 2535 HPLC connected with a 2998 PDA Detector and a 2707 Autosampler (Waters, Milford, MA, USA). Separations were performed on a COSMOSIL C18 column (5 μm, 10 × 250 mm) (Nacalai Tesque, Kyoto, Japan), a COSMOSIL C8 column (5 μm, 10 × 250 mm) (Nacalai Tesque, Kyoto, Japan) and a YMC-pack diol column (5 μm, 10 × 50 mm; 5 μm, 20 × 150 mm) (Yamamura Chemical Research, Kyoto, Japan). Direct injection high resolution ESIMS and LC-DAD-ESIMS analyses were recorded on an ultra-performance liquid chromatography-quadrupole/electrostatic field orbitrap high resolution mass spectrometry (Thermo Fisher Scientific, Waltham, MA, USA). The NMR spectra were recorded on an AVANCE III 600 MHz spectrometer (Bruker BioSpin, Ettlingen, Germany). Optical rotations were recorded on an Autopol IV Automatic Polarimeter (Rudolph Research Analytical, Hackettstown, NJ, USA).

### 3.3. Materials

The aerial part of *G. sylvestre* were collected from Nanning, Guangxi, China, in June 2019. The roots were dried at room temperature, macerated into a fine powder, and stored at room temperature. The plant was identified by Professor Songji Wei of School of Pharmaceutical Sciences, Guangxi University of Chinese medicine, Nanning, China.

### 3.4. Extraction and Isolation

The dried aerial parts of the plant (15.0 kg) were milled and then extracted with 70% EtOH (3 × 20 L, 3 days each) at room temperature to yield 1752.1 g of crude extract. Subsequently, the crude extract was suspended in H_2_O and partitioned with petroleum ether (PE) (8 × 10 L), ethyl acetate (EtOAc) (8 × 10 L), and n-butyl alcohol (n-BuOH) (8 × 10 L) to give a PE fraction (73.1 g), EtOAc fraction (383.3 g), and n-BuOH fraction (758.0 g), respectively. The EtOAc fraction (360.0 g) was subjected to HP-20 column (6 × 61 cm) chromatography eluting with a gradient solvent system of EtOH /H_2_O (10%, 30%, 50%, 70%, 80%, 95%), to yield six major fractions FA1–FA6. FA6 (75.0 g) was separated on a silica-gel column chromatography (300–400 mesh), using CH_2_Cl_2_/MeOH (100:1, 80:1, 50:1, 30:1, 20:1, 9:1, 7:1, 5:1, 3:1, 1:100, *v/v*) as a mobile phase, to obtain ten fractions FA6-1-FA6-10. FA6-5 (2.2 g) was further separated by the Sephadex LH-20 and eluted with MeOH to give six subfractions FA6-5-1–FA6-5-6. FA6-5-2 (1.9 g) by using semi-preparative HPLC (MeCN /H_2_O, 50:50–100:0, 25 min) at a rate of 4 mL/min, an injection volume of 200 μL and UV at 200 nm with column temperature at 30 °C to obtain four fractions FA6-5-2-1–FA6-5-2-4. FA6-5-2-4, by using semi-preparative HPLC [n-Hexane/n-Hexane: iso-Propyl alcohol (7:3), 40:60–0:100, 30 min] at a rate of 4 mL/min, an injection volume of 150 μL and UV at 200 nm with column temperature at 30 °C to obtain compound **1** (t_R_ = 23.10 min; 5.3 mg) and compound **2** (t_R_ = 22.04 min; 5.0 mg). Purification of fraction FA6-5-2-3 with n-Hexane/n-Hexane: iso-Propyl alcohol (7:3), 22:78, 30 min at a rate of 9 mL/min, an injection volume of 300 μL and UV at 200 nm, with column temperature at 30 °C, to afford **3** (t_R_ = 26.87 min; 5.1 mg) and **5** (t_R_ = 35.98 min; 8.9 mg) by using the YMC-pack Diol column. FA6-5-2-2, using the YMC-pack Diol column with [n-Hexane/n-Hexane: iso-Propyl alcohol (7:3), 75:25–55:45, 30 min] at a rate of 4 mL/min, an injection volume of 100 μL and UV at 200 nm, with column temperature at 30 °C, to give compound **4** (t_R_ = 15.06 min; 4.5 mg) and compound **8** (t_R_ = 16.38 min; 1.7 mg). FA6-5-2-1, by using the YMC-pack Diol column with [n-Hexane/n-Hexane: iso-Propyl alcohol (7:3), 90:1–0:100, 20 min] at a rate of 4 mL/min, an injection volume of 100 μL and UV at 200 nm, with column temperature at 30 °C, to give compound **6** (t_R_ = 16.80 min; 5.5 mg) and compound **7** (t_R_ = 15.46 min; 1.4 mg).

#### 3.4.1. Sylvepregosides A (**1**)

White powder; [α]^20^_D_ +147.3 (*c* 0.50, MeOH); UV (MeOH, nm) λ_max_ (log ε) = 230.0 (3.35), 280 (3.32) nm; IR*ν*_max_ = 3480, 2930, 1713, 1689, 1281, 1169, 1065 and 1005 cm^−1^; For ^1^H NMR (600 MHz) and ^13^C NMR (150 MHz) data can be found in Table 1; HRESIMS *m/z* 943.4423 [M + Na]^+^ (calcd for C_51_H_68_O_15_ Na: 943.4450)

#### 3.4.2. Sylvepregosides B (**2**)

White powder; [α]^20^_D_ + 145.8 (c 0.50, MeOH); UV (MeOH, nm) λ_max_ (log ε) = 210 (1.60), 280 (1.36) nm; IR*ν*_max_ = 3482, 2920, 1707, 1685, 1273, 1169, 1067, 1005 and 981 cm^−1^; For ^1^H NMR (600 MHz) and ^13^C NMR (150 MHz) data can be found in Table 1; HRESIMS *m/z* 921.4578 [M + Na]^+^ (calcd for C_26_H_31_O_7_ Na: 921.4607).

#### 3.4.3. Sylvepregosides C (**3**)

White powder; [α]^20^_D_ +56.9 (c 0.50, MeOH); UV (MeOH, nm) λ_max_ (log ε) = 215 (0.50); IR*ν*_max_ = 3441, 2940, 1701, 1681, 1236, 1082 and 983 cm^−1^; For ^1^H NMR (600 MHz) and ^13^C NMR (150 MHz) data can be found in Table 2; HRESIMS m/z 817.4339 [M + Na]^+^ (calcd for C_42_H_66_O_14_Na: 817.4350).

#### 3.4.4. Sylvepregosides D (**4**)

White powder; [*α*]^20^_D_ +32 (*c* 0.50, MeOH); UV (MeOH, nm) λ_max_ (log *ε*) = 210 (1.19); IR *ν*_max_ = 3402, 2933, 1705, 1676, 1269, 1080 and 983 cm^−1^; For ^1^H NMR (600 MHz) and ^13^C NMR (150 MHz) data can be found in Table 2; HRESIMS *m/z* 673.3553 [M + Na]^+^ (calcd for C_35_H_54_O_11_ Na: 673.3564).

### 3.5. Glucose Uptake and GLUT-4 Fusion with the Plasma Membrane

#### 3.5.1. Propagation and Maintenance of L6 Cells

For the L6 cells (Rat skeletal muscle), cell culture was procured from the American Type Culture Collection (ATCC), Manassas, Commonwealth of Virginia, America. L6 cells were cultured and maintained in α- minimum essential medium (α-MEM) with 2% inactivated fetal bovine serum (FBS) along with penicillin (100 μg/mL) and streptomycin (100 μg/mL), in a humidified atmosphere of 5% CO_2_ at 37 °C until confluent. The medium was changed every 24 h and cultured for 7 days to promote the differentiation of L6 cells into myotube cells. α-MEM, FBS, penicillin and streptomycin were obtained from Hyclone (Logan, UT, USA). The stock cultures were grown in 25 cm^2^ culture flasks, and the experiments were carried out in 60 mm petri dishes and 96-well microtiter plates. We used R-250 (Coomassie blue) staining to identify the difference between myotubes and normal L6 cells, the difference was the myotubes showed increasing cytoplasmic tonofilaments [38].

#### 3.5.2. Glucose Uptake Assay

Differentiated L6 myotubes were seeded in a 96-well black plate with the density of 1 × 10^4^–5 × 10^4^ cells/well and incubated in 100 μL α-MEM for 12 h until 100% confluence. Thereafter, L6 cells were treated with proper dosages of sample or berberine dispersed in 100 μL glucose-free α-MEM dissolved in 150 μg/mL 2-NBDG (Cayman Chemical, Ann Arbor, MI, USA). After 24 h incubation in the cell incubator, the 96-well black plate was centrifuged for 5 min at 400× *g* at room temperature. Discarding the supernatant, 200 μL cell-based assay buffer was added into each well, and then the 96-well black plate was centrifuged for 5 min at 400× *g* at room temperature. After aspirating the cell-based assay buffer, each well was added into 100 μL of assay buffer. Finally, we set the excitation/emission of the fluorescent microplate reader at 485/535 nm and analyzed fluorescence intensity of 2-NBDG in each well. Zen 2010 Software (Carl Zeiss, Jena, Germany) was used to analyze the fluorescence intensity of 2-NBDG.

#### 3.5.3. GLUT-4 Fusion with the Plasma Membrane

Construction of the myc-GLUT4-mOrange plasmid and cell line was performed as described previously [39] GLUT-4 fusion with the plasma membrane Myc-GLUT4-mOrange cells were cultured in six-well plates and grown on coverslips. After 2 h of starvation treatment, the cells were incubated in the presence of sample or berberine for 30 min. Then, cells were fixed with 3% polyformaldehyde for 30 min. After blocking with 2% bovine serum albumin (BSA; Hyclone, Logan, UT, USA) in phosphate buffered saline (PBS; Hyclone, Logan, UT, USA) for 30 min at room temperature, the cells were incubated with anti-myc mouse monoclonal antibody (TransGen Biotech, Beijing, China) for 1 h at room temperature. Then, the cells were washed three times with 2% BSA in PBS and incubated with goat anti-mouse-FITC (TransGen Biotech, Beijing, China). After being washed three times with 2% BSA in PBS, the coverslips were turned over and placed on a glass slide. Finally, mOrange red fluorescence and FITC green fluorescence were observed using a laser confocal microscope. GLUT-4 externalization was quantitated by determination of the percentage of GLUT4-mOrange-positive cells that exhibited FITC fluorescence at the cell surface.

### 3.6. Statistical Analysis

Differences between groups were analyzed by one-way analysis of variance (ANOVA). Data are shown as means ± standard error (M ± SEM). Tukey’s post hoc test of GraphPad Prism 5.0 software packages was used to perform statistical analyses. A probability (*p*) values < 0.05 were regarded as statistically significant.

## 4. Conclusions

In this study, four new C_21_ steroidal glycosides sylvepregosides A-D (**1**–**4**), as well as four known compounds (**5**–**8**) were obtained from *Gymnema sylvestre*. Compounds **1**–**6** presented the effects of glucose uptake in L6 cells at 1.10- to 2.37-fold, respectively. Specifically, compound **1** exerted the strongest activity for glucose uptake, with 1.37-fold enhancement, and compound **5** showed moderate uptake activity, by increasing glucose uptake by 0.96-fold. Further study shows that compounds **1** and **5** could promote GLUT4 fusion with the plasma membrane in L6 cells. Our research suggested that compounds **1** and **5** could offer promising lead structures with glucose uptake activity, which could be meaningful to the development of pharmaceutical products. Meanwhile, it also provided a clue for potentially active anti-diabetic constituents in the plants of genus *Gymnema.*

## Figures and Tables

**Figure 1 molecules-26-06549-f001:**
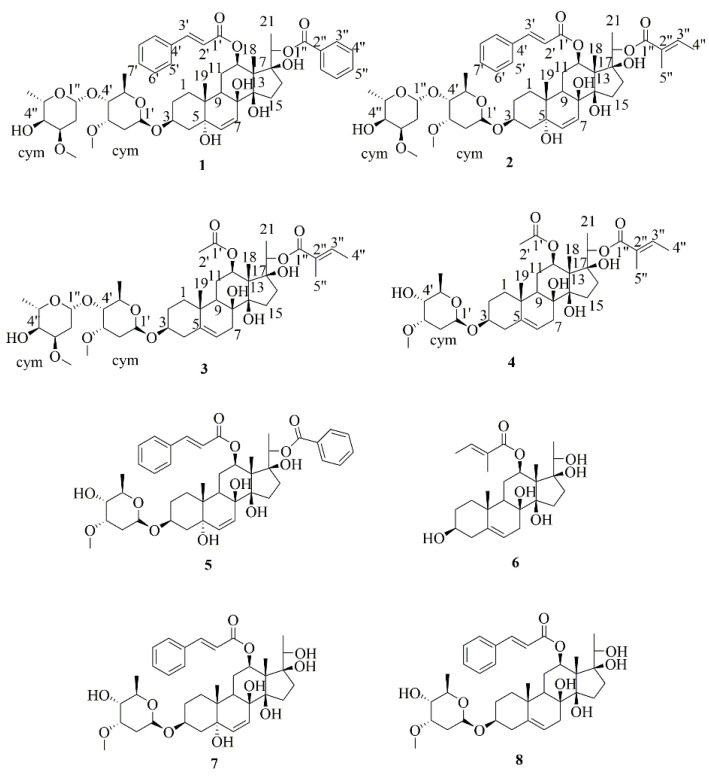
The structures of compounds **1**–**8**.

**Figure 2 molecules-26-06549-f002:**
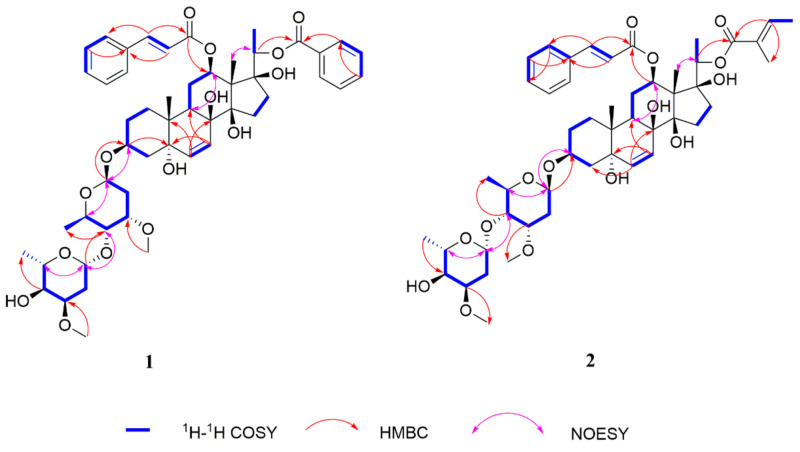
Key ^1^H-^1^H COSY and HMBC correlations of **1** and **2**.

**Figure 3 molecules-26-06549-f003:**
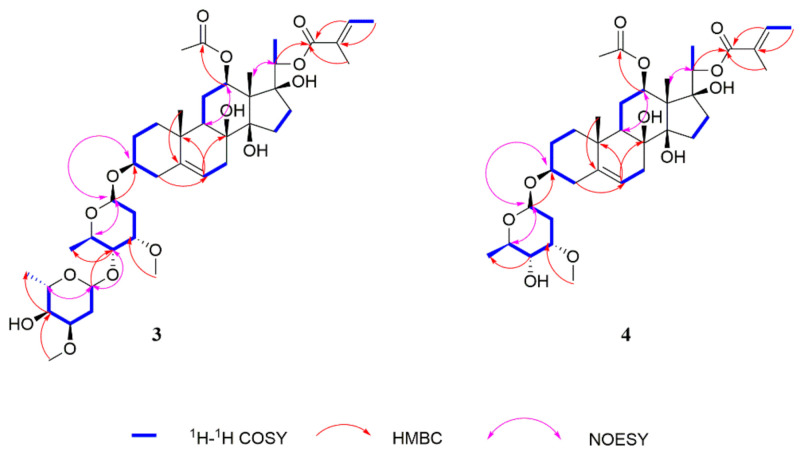
Key ^1^H–^1^H COSY and HMBC correlations of **3** and **4**.

**Figure 4 molecules-26-06549-f004:**
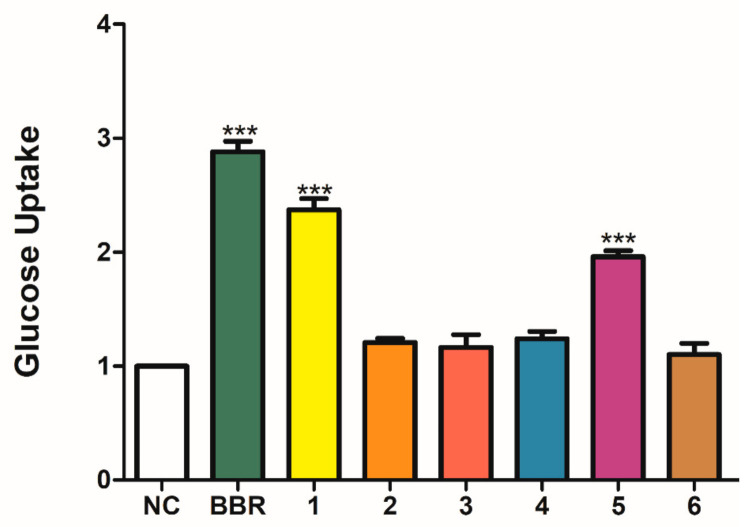
Effects of compounds **1**–**6** (30 μg/mL) on glucose uptake in L6 cells. Glucose uptake activities of compounds **1**–**6** in L6 cells using a fluorescent derivative of glucose, 2-NBDG. Berberine (30 μg/mL) was used as a positive control. After incubation for 24 h with or without 2-NBDG, the fluorescent signals were detected at Ex/Em = 485/535 nm. The results were calculated as the means ± standard error of mean (n = 5), with each performed five times; * *p* < 0.05, ** *p* < 0.01, and *** *p* < 0.001, compared to non-control group.

**Figure 5 molecules-26-06549-f005:**
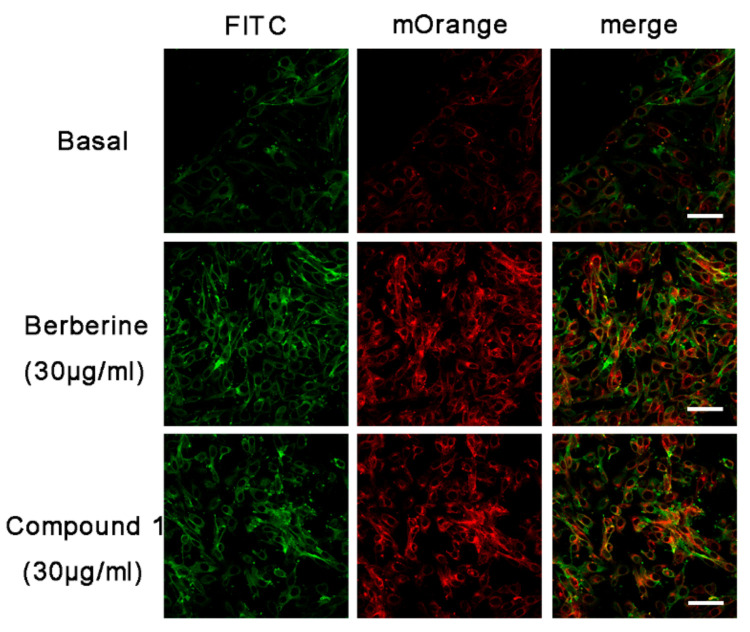
Effect of compounds **1** on fusion of GLUT4 with the plasma membrane in L6 cells. Berberine (30 μg/mL) was used as a positive control. FITC fluorescence assay in myc-GLUT4-mOrange cells treated with 30 μg/mL of compound **1** or berberine. Scale bar = 50 μm.

**Figure 6 molecules-26-06549-f006:**
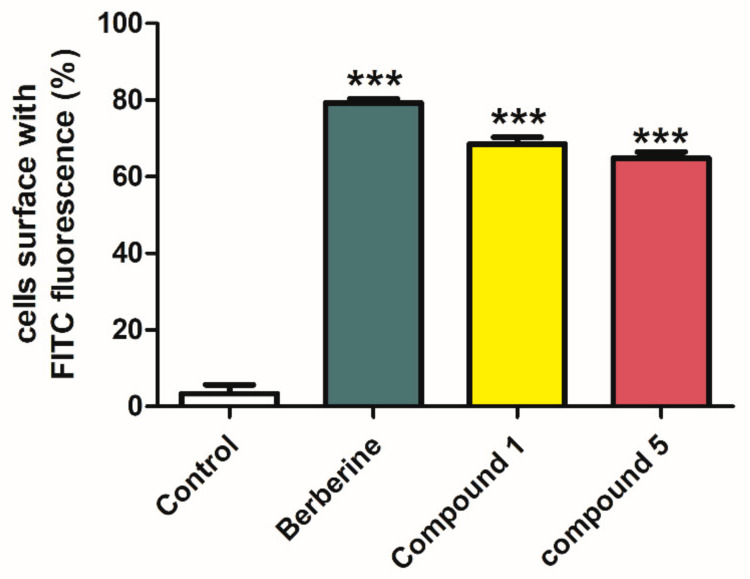
Quantitation of FITC green fluorescence cells. Treatment with 30 μg/mL of berberine or 30 μg/mL of compounds **1** and **5** for 30 min significantly increases fusion of GLUT4 with the plasma membrane in L6 cells. The results were calculated as the means ± standard error of mean (n = 3), with each performed three times; * *p* < 0.05, ** *p* < 0.01, and *** *p* < 0.001, compared to control group.

**Table 1 molecules-26-06549-t001:** ^1^H and ^13^C NMR data of compounds **1** and **2** (*δ*_H_ in ppm, *J* in Hz).

No.	1		2	
	*δ*c	*δ*_H_ mult. (*J* in Hz)	*δ*c	*δ*_H_ mult. (*J* in Hz)
1	26.1	1.53~1.56, m	26.3	1.50~1.52, m
		1.95~1.97, m		1.97~1.98, m
2	26.7	1.46~1.50, m	26.7	1.46~1.50, m
		1.77~1.80, m		1.77~1.78, m
3	74.9	4.07~4.10, m	74.9	4.07, m
4	38.5	1.6~1.62, m	38.6	1.59~1.60, m
		1.98~2.02, overlap		1.99~2.03, m
5	74.6		74.6	
6	136.7	5.63, d, (10)	136.3	5.60, d, (10.3)
7	126.3	5.83, d, (10)	126.4	5.82, d, (10.3)
8	74.3		74.1	
9	36.0	1.87~1.90, overlap	36.0	1.88~1.89, m
10	39.2		39.2	
11	22.7	1.74~1.76, m	22.8	1.74~1.76, m
		1.87~1.90, m		1.91~1.92, m
12	74.8	4.74~4.77, m	74.7	4.72, dd, (10.4, 4.4)
13	57.6		57.4	
14	87.8		88.0	
15	32.1	1.70~1.73, m	32.3	1.72~1.73, m
		1.91~1.94, m		1.89~1.91, m
16	33.5	1.98~2.02, overlap	33.1	1.93~1.95, overlap
17	87.5		87.4	
18	11.5	1.60, s	11.5	1.53, s
19	21.1	1.01, s	21.1	1.03, s
20	74.7	4.84, q, (6.1)	73.9	4.67~4.69, m
21	15.4	1.35, d, (6.2)	15.3	1.22, d, (6.2)
1′	166.6		166.5	
2′	118.9	6.07, d, (16)	119.1	6.22, d, (16)
3′	144.3	7.42, d, (16)	144.2	7.53, d, (16)
4′	134.4		134.6	
5′(9′)	128.3	7.24, d, (7.3)	128.2	7.45~7.47, m
6′(8′)	128.5	7.30~7.36, overlap	128.9	7.36~7.38, overlap
7′	130.2	7.30~7.36, overlap	130.3	7.36~7.38, overlap
1″	165.0		166.1	
2″	130.4		128.8	
3″	129.9	7.92, d, (7.2)	138.1	6.72~6.77, m
4″	128.7	7.30~7.36, overlap	14.5	1.68, d, (7.1)
5″	133.1	7.53, t, (7.4)	12.2	1.71, s
	cymⅠ		cymⅠ	
1	97.7	4.81, dd, (9.7, 1.6)	97.7	4.81, dd, (9.6, 1.8)
2	35.5	1.57~1.59, m	35.5	1.58~1.59, m
		2.11~2.14, m		2.11~2.14, m
3	77.1	3.79~3.81, m	77.1	3.80, dd, (5.8, 3.0)
4	82.2	3.24, dd, (9.7, 2.8)	82.2	3.24, dd, (9.7, 2.9)
5	68.8	3.85, dq, (9.6, 6.1)	68.8	3.85, dq, (9.6, 6.2)
6	18.2	1.21, d, (6.2)	18.2	1.20, d, (6.2)
OMe	58.2	3.45, s	58.2	3.44, s
	cymII		cymII	
1	99.5	4.67, dd, (9.7, 1.6)	99.5	4.66, dd, (10.0, 2.3)
2	33.8	1.62~1.64, m	33.8	1.62~1.63, m
		2.23~2.27, m		2.24~2.26, m
3	77.5	3.62, dd, (6.0, 2.9)	77.5	3.62, dd, (6.2, 3.1)
4	72.5	3.19, dd, (9.7, 2.8)	72.5	3.17~3.21, m
5	70.9	3.55, dq, (9.6, 6.1)	70.9	3.55, dq, (9.6, 6.2)
6	18.5	1.27, d, (6.2)	18.5	1.27, d, (6.2)
OMe	57.4	3.42, s	57.4	3.42, s

^1^H NMR and ^13^C NMR were measured at 600 and 150 MHz in CDCl_3_.

**Table 2 molecules-26-06549-t002:** ^1^H and^13^ C NMR data of compounds **3** and **4** (*δ*_H_ in ppm, *J* in Hz).

No.	3		4	
	*δ*c	*δ*_H_ mult. (*J* in Hz)	*δ*c	*δ*_H_ mult. (*J* in Hz)
1	38.9	1.06~1.10, m	38.9	1.08, dd, (13.5, 3.5)
		1.83~1.84, overlap		1.83~1.84, overlap
2	29.1	1.63~1.64, m	29.1	1.61~1.64, m
		1.89~1.93, overlap		1.87~1.92, overlap
3	78.1	3.54~3.56, m	77.9	3.53~3.56, m
4	38.9	2.29~2.32, m	38.8	2.29~2.32, m
		2.36~2.40, m		2.36~2.39, m
5	139.9		139.7	
6	118.4	5.36~5.38, m	118.6	5.36~5.37, m
7	34.6	2.14~2.17, m	34.6	2.14~2.17, m
8	74.2		74.2	
9	43.4	1.45~1.47, overlap	43.4	1.44~1.46, overlap
10	37.1		37.1	
11	24.9	1.65~1.67, m	24.9	1.64~1.66, m
		1.89~1.93, overlap		1.87~1.92, overlap
12	73.6	4.62, dd, (11.3, 4.0)	73.6	4.62, dd, (11.5, 4.2)
13	56.2		56.2	
14	88.0		87.9	
15	32.0	1.87~1.89, overlap	31.8	1.86~1.88, overlap
		1.89~1.93, overlap		1.90~1.93, overlap
16	33.2	1.87~1.89, overlap	33.2	1.86~1.88, overlap
		1.89~1.93, overlap		1.90~1.93, overlap
17	87.9		88.1	
18	10.4	1.44, s	10.4	1.44, s
19	18.3	1.13, s	18.3	1.14, s
20	74.2	4.65~4.66, m	74.2	4.65, q, (6.2)
21	15.1	1.23, d, (5.6)	15.1	1.22, d, (6.1)
1′	166.4		166.4	
2′	128.8		128.8	
3′	138.0	6.83~6.87, m	138.1	6.83~6.87, m
4′	14.7	1.82, d, (7.0)	14.8	1.81, d, (7.1)
5′	12.3	1.85, s	12.3	1.85, s
1″	171.3		171.3	
2″	21.9	1.94, s	21.9	1.94, s
	cymI		cym	
1	96.2	4.84, dd, (9.7, 1.7)	95.7	4.78, dd, (9.6, 1.8)
2	35.6	1.55~1.59, m	34.2	1.57~1.60, m
		2.09~2.12, m		2.19~2.22, m
3	77.2	3.81, dd, (5.9, 2.8)	77.6	3.62, dd, (6.1, 3.0)
4	82.6	3.22, dd, (9.7, 2.9)	72.6	3.21, dd, (9.5, 3.6)
5	68.6	3.85, dq, (9.6, 6.1)	70.9	3.57, dq, (9.7, 6.1)
6	18.3	1.22, d, (6.0)	18.4	1.27, d, (6.2)
OMe	58.1	3.44, s	57.4	3.43, s
	cymII			
1	99.5	4.67, dd, (9.5, 1.8)		
2	33.8	1.60~1.62, m		
		2.23~2.26, m		
3	77.5	3.62, (6.2, 3.1)		
4	72.5	3.19, dd, (9.7, 3.6)		
5	70.8	3.55, dq, (9.5, 6.2)		
6	18.5	1.27, d, (6.2)		
OMe	57.4	3.42, s		

^1^H NMR and ^13^C NMR were measured at 600 and 150 MHz in CDCl_3_.

## Data Availability

Not available.

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
