# Peer review of "Identification of C21 Steroidal Glycosides from Gymnema sylvestre (Retz.) and Evaluation of Their Glucose Uptake Activities"

_molecules, 2021, doi:10.3390/molecules26216549_

Round 1

Reviewer 1 Report

Interesting article. Please add melting points for new compounds. Should be accepted after a few corrections. In some places spaces are missing, be careful and consequently write G. sylwstrii in italics.

Author Response

Point 1: Please add melting points for new compounds. Should be accepted after a few corrections. In some places spaces are missing, be careful and consequently write G. sylwstrii in italics.

Response: Thank you for your advices. The new compounds in this experiment are powder and have no fixed melting point, so it is difficult to determine their melting point.

Reviewer 2 Report

Dear Authors,

Congratulations for the extensive and well-made characterization of the compounds extracted from the G. sylvestre fraction. I suggest that further experiments should be carried out to confirm the anti-diabetic activity of these compounds in the future.

Best regards.

Author Response

Point 1: Congratulations for the extensive and well-made characterization of the compounds extracted from the G. sylvestre fraction. I suggest that further experiments should be carried out to confirm the anti-diabetic activity of these compounds in the future.

Response: Thank you for your advices. We have supplement the experiment of GLUT4 fusion with the plasma membrane to further confirm the anti-diabetic activity of these compounds.

Reviewer 3 Report

Very interesting paper. I can be published in presented form. One important paper is missing and should be discussed.

Kumar, P.M. et al. (2016) Methanolic leaf extract of Gymnema sylvestre augments glucose uptake and ameliorates insulin resistance by upregulating glucose transporter-4, peroxisome proliferator-activated receptor-gamma, adiponectin, and leptin levels in vitro. J Intercult Ethnopharmacol. 5(2): 146–152.

Author Response

Very interesting paper. I can be published in presented form. One important paper is missing and should be discussed.

Kumar, P.M. et al. (2016) Methanolic leaf extract of Gymnema sylvestre augments glucose uptake and ameliorates insulin resistance by upregulating glucose transporter-4, peroxisome proliferator-activated receptor-gamma, adiponectin, and leptin levels in vitro. J Intercult Ethnopharmacol. 5(2): 146–152.

Response: Thank you very much for your good advices. We have discussed the above paper in Lines 51-55.

Reviewer 4 Report

Review Comments: molecules-1378917

Comments to Authors

Liu et al. report the article "Identification of C21 steroidal glycosides from Gymnema sylvestre (Retz.) and evaluation their glucose uptake activities." As shown in the manuscript, the authors wanted to discover the potential glucose uptake active composition of G. sylvestre. In this research, the new C21 steroidal glycosides sylvepregosides A and gymnepregoside H with the glucose uptake active from G. sylvestre. Overall, this manuscript was a phytochemistry research report. However,  it was only used one screen method but still could not support enough information for pharmacological research.

Comments

  1. More than hundreds of reports about Gymnema sylvestre or its derivate compounds (for example, gymnemic acid; dihydroxy gymnemic triacetate; conduritol A) influence glucose metabolism or treatment for diabetes. However, the authors did not cite and described them well. Suggest the authors must explain them well.
  2. Methods didn't include the cell-cultured, the glucose uptake activity assay, and Statistical analysis. Suggest the authors must supplement the methods well.
  3. The authors isolated eight compounds from Gymnema sylvestre. Why only detected six compounds for the glucose uptake activities assay in this manuscript? Suggest the authors must explain them well.
  4. The authors did not supplement any could double-check methods for pharmacological research in this manuscript. Suggest the authors must support them well.
  5. In Figure 4, there is no dosage-dependent information for each compound. What could the authors detect the efficacy of the eight compounds? Suggest the authors must explain them well.
  6. In Figure 4, there are no pieces of information about positive control BBR. Why did the authors use BBR as the positive control but not the derivate compounds of Gymnema sylvestre (gymnemic acid; dihydroxy gymnemic triacetate; Conduritol A)? Suggest the authors must explain them well.

Round 2

Reviewer 4 Report

The comments from the reviewer: molecules-1378917

The authors already correct the revised manuscript of “Identification of C21 steroidal glycosides from Gymnema sylvestre (Retz.) and evaluation their glucose uptake activities”. However, all the figure legends that only showed the title still made the results unclearly presented. Suggest the authors must write the effective figure legends in this manuscript.

Author Response

Thank you for your advices.  We have added new figure legends in the manuscript.